# Epigenetic Modulation of Opioid Receptors by Drugs of Abuse

**DOI:** 10.3390/ijms231911804

**Published:** 2022-10-05

**Authors:** Ke Zhang Reid, Brendan Matthew Lemezis, Tien-Chi Hou, Rong Chen

**Affiliations:** 1Department of Biology, Center for Molecular Signaling, Wake Forest University, Winston-Salem, NC 27109, USA; 2Department of Physiology and Pharmacology, Wake Forest University School of Medicine, Winston-Salem, NC 27157, USA

**Keywords:** opioid receptors, gene expression, drugs of abuse, DNA methylation, histone modifications, noncoding RNAs, epigenetics

## Abstract

Chronic exposure to drugs of abuse produces profound changes in gene expression and neural activity associated with drug-seeking and taking behavior. Dysregulation of opioid receptor gene expression is commonly observed across a variety of abused substances including opioids, cocaine, and alcohol. Early studies in cultured cells showed that the spatial and temporal gene expression of opioid receptors are regulated by epigenetic mechanisms including DNA and histone modifications and non-coding RNAs. Accumulating evidence indicate that drugs of abuse can modulate opioid receptor gene expression by targeting various epigenetic regulatory networks. Based on current cellular and animal models of substance use disorder and clinical evidence, this review summarizes how chronic drug exposure alters the gene expression of mu, delta, kappa, and nociceptin receptors via DNA and histone modifications. The influence of drugs of abuse on epigenetic modulators, such as non-coding RNAs and transcription factors, is also presented. Finally, the therapeutic potential of manipulating epigenetic processes as an avenue to treat substance use disorder is discussed.

## 1. Introduction

Substance use disorder (SUD) is characterized by excessive and continued drug use that disrupts social and occupational activity [1]. According to the National Center for Drug Abuse Statistics, the number of drug overdose deaths in the United States has increased by about 30% yearly since 1999. Chronic exposure to abuse drugs produces profound changes in gene expression and neural activity in various brain regions, which contributes to persistent drug-seeking and taking behavior. Although different classes of drugs exert their action on specific targets, dysregulation of the opioid system, particularly of opioid receptors, is commonly observed across a wide variety of substances, including opioids, cocaine, methamphetamine, alcohol, and nicotine [2]. Changes in the expression of opioid receptors have been associated with increased vulnerability to drug-seeking and taking behavior in both human and animal models of SUD [3,4]. Early studies in cultured cells demonstrated that opioid receptors are subject to epigenetic regulations [5]. Since it has become increasingly appreciated that drugs of abuse modify epigenetic regulation in the brain [6,7], this review summarizes the mechanisms underlying epigenetic modulations of opioid receptor subtypes in the context of SUD, which has not been reviewed previously. We also discuss the potential of applicability of drugs, which can target epigenetic regulatory events, as an avenue for the treatment of SUD.

## 2. The Physiology and Pharmacology of Opioid Receptors

There are four types of opioid receptors: mu-opioid receptors (MORs), delta-opioid receptors (DORs), kappa-opioid receptors (KORs), and nociceptin/orphanin FQ receptors (NOP receptors). They are class A rhodopsin-like G-protein-coupled receptors that contain seven transmembrane domains, an extracellular N-terminus and an intracellular C-terminus, and are primarily coupled to heterotrimeric Gαi/o proteins [6]. Opioid receptors share a high sequence homology (73–100%) within transmembrane domains but differ significantly in the N- and C-terminus with only 9–20% sequence homology [6,7,8,9,10]. The NOP receptors are closely related but occupy a non-opioid branch of the opioid family of receptors [11]. The endogenous ligands for MORs and DORs are endorphins and enkephalins, and dynorphins are the primary endogenous ligands for KORs [12]. Nociceptin or orphanin FQ (N/OFQ) are endogenous ligands for NOP receptors [13]. There are also natural and synthetic opioids, such as morphine, heroin, fentanyl, oxycodone, and codeine, which differ in their binding affinities for each receptor subtype [14,15].

MORs and DORs are distributed in various brain regions, the dorsal horn of the spinal cord, and peripheral tissues based on in situ hybridization mRNA analysis [16,17]. MORs and DORs share approximately 60% amino acid sequence homology in mice and, therefore, their individual ligands often show a binding affinity for both receptors [18]. The analgesic effect of opioids arises from the activation of MORs and, to a lesser extent, of DORs in the dorsal horn of the spinal cord [19,20]. MORs and DORs are also abundantly expressed in the mesolimbic and mesocortical pathways that are associated with reward. The expression levels of MORs and DORs influence behavioral responses to drugs. Mice with MOR deletion exhibit reduced morphine and cocaine self-administration, attenuated alcohol drinking in two-bottle choice, and disrupted nicotine-induced conditioned place preference [21,22,23,24,25]. Likewise, deletion of DOR in mice leads to disturbed conditioned place preference for morphine, less motivation to acquire cocaine, and decreased nicotine self-administration [26,27,28]. However, DOR knockout mice show increased alcohol drinking in two-bottle choice [29,30], which is in contrast to the observation in MOR knockout mice.

KORs are the most abundant opioid receptor type in the brain [31] and play an important role in the regulation of both reward and mood processes [32]. For example, mice with KOR deletion show attenuated morphine withdrawal effect and reduced alcohol drinking in two-bottle choice [29,33,34]. Mice with conditional KOR knockout in dopaminergic neurons display enhanced sensitivity to cocaine-induced locomotor stimulation but reduced anxiety-like behavior [35]. Moreover, conditional knockout of KOR in the basolateral amygdala blocks stress-induced conditioned place preference for nicotine [36]. These data indicate that KORs may play an important role in comorbidity of substance use disorder and anxiety-like behavior.

NOP receptors are distributed throughout the central and peripheral nervous systems but are highly expressed in the mesocorticolimbic systems, where they modulate the transmission of certain neurotransmitters including dopamine, glutamate, and GABA [37,38,39]. They have a distinct pharmacological profile from that of other opioid receptor subtypes, which is characterized by a low affinity for classical opioid peptides and antagonists [11]. Activation of NOP receptors produces dichotomous responses to pain that are site-specific. For example, administration of orphanin FQ induces anti-nociceptive responses, whereas intracerebroventricular injection of OFQ blocks opioid-induced analgesia [40]. Moreover, the expression of NOP receptors is implicated in drug reward. To illustrate, mice lacking NOP receptors show enhanced cocaine-induced locomotor sensitization [41]. Further, absence of NOP receptors in rats results in decreased motivation to self-administer nicotine, reduced nicotine intake, and attenuated cue-induced nicotine reinstatement [42]. Additionally, NOP receptor knockout mice consume less alcohol in a binge-like alcohol drinking paradigm [43].

To summarize, the expression level of each opioid receptor subtype influences the responses to drug reward and analgesics. Importantly, prior studies have shown that the gene expression of these opioid receptors is altered following chronic exposure to drugs of abuse including cocaine [44], oxycodone [45], nicotine [46], and alcohol [47]. Therefore, it is critical to understand how drugs of abuse alter the gene expression of opioid receptor subtypes so that better pharmacotherapies could potentially be developed for the treatment of SUD.

## 3. Epigenetics and SUD

Epigenetics denote persistent changes in gene expression without an alteration in DNA sequence. The epigenetic modification of gene expression can be continued even after the initial causative stimuli have been removed or can be inherited for at least one successive generation. Epigenetic mechanisms trigger conformational changes to chromatin, which controls the accessibility of the DNA template for transcription [48,49]. The state of chromatin is regulated by biochemical processes, including DNA modifications, post-translational modifications of histones, noncoding RNAs (ncRNAs), and chromatin remodeling (Figure 1).

SUD meets an essential criterion of epigenetic regulation: continued influence of drugs long after the drug has been removed from the system. Repeated drug exposure may cause a long-lasting alteration of the transcriptional program, leading to enduring transcriptional changes of genes implicated in neurotransmission and, therefore, persistent drug-seeking and taking behavior. Recent studies have shown that manipulation of various aspects of epigenetic processes can impact behavioral responses to drugs. For instance, microinjection of a DNA methyltransferase (DNMT) inhibitor into rat nucleus accumbens (NAc) to reduce DNA methylation suppresses cue-induced cocaine-seeking behavior, whereas microinjection of the methyl donor S-adenosylmethionine to increase DNA methylation produces the opposite effect [50]. Systemic injection of a histone deacetylase inhibitor to increase histone acetylation enhances rat striatal histone H3 hyperacetylation and the reinforcing effect of cocaine, assessed by conditioned placed preference [51]. Furthermore, overexpression of microRNA-495, a small ncRNA, in mouse NAc suppresses the motivation to seek and take cocaine [52]. Notably, drug-induced transgenerational epigenetic inheritance has been reported. A study of the male offspring of female rats that had repeated exposures to morphine found that the males displayed increased anxiety-like behavior during adolescence, increased sensitivity to the analgesic effects of morphine, and developed a greater tolerance to chronic morphine treatment [53]. These findings indicate that exposure to drugs of abuse can produce epigenetic changes through various mechanisms. These epigenetic changes are associated with the reinforcing and rewarding properties of drugs and, hence, may serve as potential targets for developing new therapeutic interventions.

## 4. Drugs of Abuse Modify Epigenetic Regulation

### 4.1. DNA Methylation

DNA methylation occurs via the addition of a methyl group to a cytosine residue within a cytosine-phosphate-guanine (CpG) site on the 5′ carbon (5-mC) of the pyrimidine ring. Regions with a high frequency of CpG sites, termed CpG islands, are heavily enriched in the promoter regions of housekeeping genes [54,55]. The CpG-rich promoters contain an elevated G/C content and often lack the core promoter elements, such as a TATA-box [56]. In mammals, CpG methylation (5-mC) at a promoter is commonly correlated with gene silencing [57,58]. The methylation is catalyzed by DNA methyltransferases, including DNMT1, DNMT3A, and DNMT3B (Figure 1A) [59]. Different forms of DNA modifications are generated by removing 5-mC via Ten-Eleven Translocation (TET) family enzymes, which are α-ketoglutarate (α-KG)/Fe(II)-dependent dioxygenases [60]. The 5-mC can be oxidized to 5-hydroxymethyl (5-hmC), 5-formyl (5-fC), and 5-carboxyl (5-caC) [60]. Among the forms of cytosine modifications, 5-hmC is most abundant in the brain and enriched in gene bodies, suggesting that it may play a particular role in gene activation [61,62]. Moreover, oxidation of 5-mC to 5-hmC is necessary for mammalian neuronal differentiation and function [63]. Accumulating evidence indicates that drugs of abuse influence DNA methylation in the brain. For example, chronic morphine treatment (10 mg/kg, twice/day, 9 days) to rats alters 5-mC and 5-hmC levels in a brain region-specific manner [64]. Further, global 5-mC levels are increased in rat NAc following withdrawal from chronic cocaine treatment, and this hypermethylation is associated with gene repression [50].

The TET family (TET1-3) proteins participate in the conversion of 5-mC to 5-hmC (Figure 1A). They are highly expressed in the brain, with TET3 being most abundantly expressed in the cerebellum, prefrontal cortex, and hippocampus compared to TET1 and TET2 [65]. Drugs of abuse alter the expression of DNA modifying enzymes. For example, acute cocaine treatment decreases DNMT1, DNMT3a, TET1, and TET2 transcript levels in mouse NAc; however, the DNMT transcripts and enzyme activity levels are increased after a 24 h withdrawal period from chronic cocaine treatment [66]. Further, cocaine-induced behavioral sensitization is associated with increased expression and activity of DNMTs and decreased expression and activity of TET1 and TET3 in mouse NAc [66]. These changes are correlated with altered 5-mC and 5-hmC levels at the candidate gene promoter regions.

### 4.2. Histone Modifications

Post-translational modifications of histones regulate the structure and activity of chromatin. The N-terminal histone tails are subject to modifications including, but not limited to: acetylation, methylation, O-GlcNAcylation, ubiquitination, phosphorylation, SUMOylation, and serotonylation [67,68]. Histone modifying enzymes, broadly defined as writers or erasers, can add or remove modifications, respectively. Moreover, readers are another class of proteins that recognize their cognate epigenetic marks and initiate changes in chromatin structure and the binding of TFs at specific loci [69]. The function of each epigenetic mark can be dichotomously categorized into either gene activation or repression (Figure 1B).

Histone acetylation via histone acetyltransferases (HATs) is often associated with transcriptional activation, while deacetylation via histone deacetylases (HDACs) is associated with transcriptional inhibition. Histone acetylation and deacetylation in the brain have increasingly been reported following exposure to drugs [70,71]. For instance, analyses of postmortem brains of human heroin users show hyperacetylation of lysine 27 of histone H3 (H3K27ac) in the striatum [72]. Moreover, acetylated histone H3 levels positively correlate with the length of drug use. Further investigation of chromatin accessibility using transposase-accessible chromatin sequencing reveals that H3K27 acetylation (H3K27ac) generates an open chromatin state. Since this modification is often found at enhancer regions [73,74], H3K27ac may induce gene activation following chronic exposure to heroin, as observed in human heroin users [72]. In addition, inhibition of HDAC activity using sodium butyrate increases heroin reinstatement in rats [75]. Collectively, these data suggest that chronic exposure to drugs increases transcriptional activities, which, in turn, may lead to heightened drug-seeking and taking behavior [70,71,76].

Histones can also be methylated at multiple lysine or arginine residues. Unlike histone acetylation, which is generally associated with gene activation, histone methylation is involved in both transcriptional activation and silencing depending on the deposition of the methyl group(s). For example, histone H3 lysine 4 trimethylation (H3K4me3) is frequently found at the promoter regions of transcriptionally active genes [77]. However, methylation of histone H3 lysine 9 (H3K9me) and lysine 27 (H3K27me) mark the promoters of transcriptionally silent genes [78]. These repressive histone modifications lead to chromatin compression and, thus, suppression of transcription at these loci. To date, very few have studied how drugs of abuse alter histone methylation compared to histone acetylation. Nonetheless, currently available studies implicate that chronic drug exposure alters histone lysine or arginine methylations, which play essential roles in regulating gene expression. For instance, repeated systemic injections of cocaine reduce global levels of H3K9me2 in mouse NAc, leading to a more transcriptionally active state of chromatin [79]. In rodents and humans, repeated cocaine exposure significantly decreases Protein Arginine Methyltransferase-6 (PRMT6) and its associated histone mark, asymmetric dimethylation of R2 on histone H3 (H3R2me2a, a repressive mark) in the NAc. This histone mark has been shown to protect against cocaine self-administration and conditioned place preference for cocaine [80]. Consistent with enhanced transcriptional activity, chronic exposure to drugs reduces transcriptional silencing of histone methylations, such as H3K9me and H3R2me2a, whereas it enhances histone methylation marks that are associated with transcriptional activation such as H3K4me3 [81,82,83]. Although ample progress has been made toward understanding the influence of drugs of abuse on histone acetylation and methylation, it is worth noting that many other histone modifications, such as O-GlcNAcylation, SUMOylation, ubiquitination, etc., have yet to be investigated in the context of SUD.

### 4.3. ncRNAs

ncRNAs play essential roles in chromatin remodeling and post-transcriptional gene silencing. Unlike housekeeping ncRNAs, which are the products of RNA polymerase I & III, the ncRNAs described here are transcribed by RNA polymerase II but are not translated into proteins. Based on their length, ncRNAs are divided into two categories: long non-coding RNAs (lncRNAs), when longer than 200 bp, and small non-coding RNAs (sncRNAs), when shorter than 200 bp [84]. lncRNAs are produced from overlapping protein-coding genes, fragments of introns, anti-sense transcripts, enhancers, 5′ or 3′ untranslated regions, or repetitive DNA elements, including transposons and their remnants [85]. Over the past decade, there have been intense debates regarding whether pervasive lncRNAs are functional. The current view is that most lncRNAs generated from excess transcription are non-functional; however, a number of other lncRNAs do have specific functional roles, for example, in chromatin remodeling (Figure 1C) [85].

Micro RNAs (miRNAs) are sncRNAs that are generally 22 nucleotides in length and inhibit mRNA translation by degrading or destabilizing their mRNA targets [86]. Several reports indicate that drugs of abuse alter the expression of certain sncRNAs (Figure 1D). For instance, Argonaute 2 (Ago2) protein plays an essential role in miRNA generation and miRNA-mediated gene silencing [87]. Ablation of *Ago2* in mouse striatal dopamine D2 receptor-containing neurons decreases the reinforcing effect of cocaine measured by conditioned place preference and motivation to self-administer cocaine [87]. This study suggests that Ago2-dependent miRNAs play an important role in regulating the rewarding and reinforcing properties of cocaine. In recent years, more miRNAs responsive to drugs of abuse have been identified. Morphine or fentanyl treatment increases microRNA 339-3p (miR-339-3p) in cultured primary mouse hippocampal neurons [88]. Further, both acute and chronic treatment of methamphetamine (METH) change the expression of specific miRNAs in rat NAc [89]. Notably, the levels of several miRNAs, including miR-496-3p, miR-194-5p, miR-200b-3p and miR-181a-5p are significantly altered in rats chronically, but not acutely, treated with METH, suggesting that these miRNAs may be associated with METH-induced persistent adaptations in the brain [89]. Analysis of blood samples from abstinent METH-dependent patients shows an increase in miR-143 levels and this increase is associated with METH-induced disruption of the blood-brain barrier [90]. Many other miRNAs including miR-181, miR-212, miR-124, miR-9, and let-7 have also been identified in animal models of SUD [91,92,93,94].

lncRNAs are also involved in gene expression, RNA stabilization, and chromatin remodeling. Evidence from clinical data and animal models of SUD indicate that drugs of abuse also change the expression of lncRNAs. For example, lncRNAs are upregulated in patients with a history of cocaine or heroin use [95,96]. Chronic METH treatment (IP, 2 mg/kg, 5 days) alters the lncRNA profiles in mouse NAc [97]. Moreover, prolonged cocaine exposure (10 µM, 24 h) to mouse hippocampal HT22 cells increases the expression of Maternally Expressed Gene 3 (MEG3), which is a lncRNA; this upregulation is also associated with morphine-induced autophagy [98]. In addition, several single nucleotide polymorphisms (SNPs) located at lncRNA loci are associated with SUD. Genetic variants within ANRIL (antisense non-coding RNA in the INK4 locus) are associated with an increased risk of METH-dependence in humans [99]. A SNP in the 3′ untranslated region (3’ UTR) of the circadian Vasoactive Intestinal Peptide Receptor 2 (VIPR2) gene (*VIPR2* SNP rs885863) is significantly noted in a pool of patients suffering from opioid dependence [100]. This SNP is an expression quantitative trait locus for *VIPR2* and a long intergenic non-coding RNA, lncRNA 689. Lastly, circular RNAs (circRNAs) represent a large class of ncRNAs that have emerged as key regulators of gene expression in recent years. The circRNAs can repress the activities of corresponding miRNAs through direct binding and are significantly enriched in the brain [101]. It has been reported that cocaine self-administration influences the striatal expression of ninety different circRNAs [102]. To date, the role of circRNAs and the molecular mechanisms by which they modulate gene expression in SUD remain largely unknown and warrant future investigation.

In summary, chronic drug exposure alters the expression of enzymes involved in modifying DNA or histones in the brain [103]. These effects are associated with prominent changes in the patterns and the amount of DNA or histone modifications on the promoters or gene bodies. These epigenetic changes perhaps underlie the long-lasting neurochemical and behavioral adaptations observed in human and animals following chronic exposure to drugs of abuse.

## 5. Epigenetic Modulation of Opioid Receptor Gene Expression by Drugs of Abuse

In humans and rodents, each opioid receptor is encoded by a single gene [6]. The coding regions for MOR, DOR, and KOR are remarkably conserved at the 7 transmembrane regions, not only at the DNA and amino acid sequence levels but also at their splicing junctions [6]. Nonetheless, their gene sequences and genomic structure outside the transmembrane domains differ substantially, particularly at the 5′ and 3′ UTRs, thereby producing multiple mRNA variants [104,105]. The promoters for opioid receptor subtype genes are mostly TATA-less and can start transcription from multiple sites. They also use common TFs for transcription. A large body of research on the epigenetic regulation of opioid receptors has been conducted in cultured cells. Recent accumulating evidence indicate that the gene expression of opioid receptors in the brain is impacted by a variety of drugs of abuse (Table 1). This section primarily focuses on how opioid receptors are epigenetically modulated based on currently available data from human drug users and animal models of SUD.

### 5.1. Epigenetic Modulation of the MOR Gene

#### 5.1.1. MOR Gene

The gene encoding MOR (*OPRM* or *Oprm*) can use both canonical and alternative transcription and translation initiation sites from mRNA variants that differ in their 5′ UTR [106,107,108,109]. For instance, mouse *Oprm1* is located in chromosome 10 [110], and uses two promoters nearby: the distal promoter and the proximal promoter (Figure 2A). The distal promoter guides the initiation of a major transcript, while the proximal promoter drives the production of multiple transcripts with different translation initiation sites [6]. Additionally, the MOR transcript undergoes extensive alternative splicing to produce numerous splice variants in a pattern that is conserved from rodents to humans. These splice variants are classified into three categories: C-terminal splicing variants, 6 transmembrane (TM) domain variants, and single TM1 variants [111]. The variants resulting from the C-terminal splicing are expressed in a brain region-specific manner. Notably, these variants diverge in agonist-induced G protein coupling, phosphorylation, internalization, recycling, degradation, and response with respect to morphine exposure [6]. Moreover, a recently identified splice variant in mice that lacks the first TM domain appears to form a heterodimer with the full-length MORs, resulting in enhanced expression of the full-length receptors [112]. The truncated single-TM variants do not bind to any opioids; instead, they promote morphine analgesia by working as molecular chaperones to stabilize the non-truncated MOR [113]. It is important to note that the functions of these splice variants in vivo are essentially unknown.

#### 5.1.2. MOR Gene Expression, DNA Methylation, and Drugs of Abuse

Mouse *Oprm1* promoters are TATA-less but contain abundant GC-enriched CpG islands that can be heavily methylated, which leads to gene silencing. For instance, the *Oprm1* gene is completely suppressed in mouse embryonal carcinoma P19 cells; however, there is a robust increase in *Oprm1* expression when the cells are differentiated by retinoic acid [155]. The elevated expression levels are correlated with the degree of hypo-methylation at the proximal promoter region of *Oprm1*. The rise and fall of *Oprm1* expression during cell differentiation mirror the effect of treatment with either HDAC or DNA methylation inhibitors on *Oprm1* expression [155]. DNA methylation at the *Oprm1* promoter regions contributes to the low expression of MORs in the brain [156].

The human *OPRM1* promoter is hypermethylated following acute and chronic exposure to opiates. To illustrate, in a genome-wide DNA methylation study of saliva samples from opioid-naive dental surgery patients, nine out of ten selected CpG sites in the *OPRM1* promoter show elevated methylation in patients approximately 40 days after the surgery following higher doses of morphine treatment [157]. Furthermore, patients with a prior history of heroin use and that were on methadone maintenance treatment show increased methylation at two CpG-rich islands in peripheral lymphocytes [158]. The methylated sites are located at the predicted Sp1 transcription factor-binding sites and may prevent Sp1 and other transcriptional activators from accessing the locus, resulting in a low expression of *OPRM1* as noted in the lymphocytes of these patients [158]. A functional genetic variant N40D (*OPRM1* 118A>G, rs1799971) observed in chronic opiate users results in a new CpG-methylation site within the *OPRM1* locus, which enhances DNA methylation and leads to gene silencing from this site forward [159]. The hypermethylation at this site is associated with reduced *OPRM1* mRNA transcription and receptor expression in the postmortem brains of these opioid users. Increased DNA methylation is also reported at an additional CpG-rich island of the *OPRM1* promoter in leukocytes of prior opiate users; however, DNA methylation at this site does not induce a change in the transcriptional level of *OPRM1* [160]. In addition, there is an increase in methylation at the long interspersed nuclear elements 1 (LINE-1), a global methylation site, in leukocytes of these patients, and this increase is strongly correlated with an increased pain experience [160]. Interestingly, elevated DNA methylation at the *OPRM1* promoter is also reported in the spermatozoa of humans with a history of opioid use, suggesting an intergenerational epigenetic inheritance of chronic opioid-induced DNA hypermethylation [161]. Furthermore, following opioid exposure, hypermethylation within the *OPRM1* promoter is repeatedly observed in infant’s cord blood or saliva with neonatal abstinence syndrome, suggesting the likelihood of *OPRM1* gene suppression in the offspring [162]. Collectively, these data suggest that drugs of abuse can stimulate DNA methylation of the *OPRM1* gene.

A potential mechanism underlying altered DNA methylation induced by drugs is an upregulation of MeCP2, a nuclear protein that reads and binds to methylated CpG dinucleotides and subsequently recruits HDACs to silence the gene (Figure 3A). A few studies have suggested an association between MeCP2 function and *Oprm1* expression. In P19 cells, DNA methylation levels are positively associated with MeCP2 binding at the *Oprm1* promoter region [155]. Moreover, MeCP2 expression levels are inversely correlated with *Oprm1* levels in P19 cells [155]. Therefore, MeCP2 may control *Oprm1* expression via binding to the methylated CpG islands at the receptor promoter region. This notion is further supported by a study showing that surgical injury to mouse DRG results in hypermethylation of the *Oprm1* gene promoter, as well as increased expression of MeCP2 [163]. MeCP2 knockdown rescues the expression of *Oprm1* and restores the analgesic effect of morphine, suggesting a direct causal effect of MeCP2 on *Oprm1* expression. Moreover, MeCP2 can bind to and repress G9a, also known as Ehmt2, a histone methyltransferase that writes the repressive mark H3K9me2 [164]. Thus, it appears that MeCP2 can epigenetically modulate *Oprm1* expression through both DNA methylation and histone modification.

In addition to MeCP2, methyl-CpG-binding domain (MBD) proteins can silence genes by binding to methylated CpG sites and recruiting other transcriptional corepressors (e.g., HDACs) to the promoter region of the targeted genes [166]. Among the MBD proteins, MBD1 proteins repress *Oprm1* gene expression by recruiting DNMT3a to the *Oprm1* promoter [167]. As a result, *Mbd1* knockout mice exhibit reduced responses to acute noxious stimuli and blunted neuropathic pain, which is rescued by overexpression of the *Mbd1* gene [167]. Altogether, these data suggest that drugs of abuse may affect MOR gene expression by altering DNA methylation at its promoter regions or by affecting the binding of its downstream effector proteins.

#### 5.1.3. MOR Gene Expression, TFs, and Drugs of Abuse

Numerous TFs have been identified that work to regulate *Oprm1* expression positively or negatively to fine-tune its expression in various cell culture models. However, there are only a few studies on TFs that modulate *Oprm1* expression in the context of SUD. The cAMP response element (CRE)-binding protein (CREB) positively regulates *Oprm1* expression. At the 5′ UTR region, CRE is located at -106/-111 of *Oprm1* [142]. In rat PC-12 cells, Forskolin-stimulated activation of protein kinase A (PKA) leads to the binding of CREB and CREB-binding protein (CBP) to the *Oprm1* promoter and increases *Oprm1* expression [142], suggesting a cAMP-mediated pathway for *Oprm1* gene modulation. Interestingly, fentanyl treatment (10 ng/mL) to PC-12 cells induces a time-dependent (1–48 h) increase in *Oprm1* expression, while morphine (1 µg/mL) has no effect [142]. The fentanyl-mediated increase in *Oprm1* expression is abolished by a PKA inhibitor, H89, indicating that fentanyl may modulate *Oprm1* expression by promoting the binding of CREB and CBP to the promoter of *Oprm1*. Although both fentanyl and morphine are opioid analgesics that bind to MORs, fentanyl shows a greater analgesic effect, a higher abuse potential, and more severe respiratory depression than morphine does [168,169]. Moreover, fentanyl produces less tolerance than morphine [170] and has milder side effects of nausea and itching [171]. The upregulation of *Oprm1* expression by fentanyl, but not by morphine, may explain the observation that repeated exposure to fentanyl, but not to morphine, produces minimal tolerance, as well as other effects.

A morAP-2-like element is present at the mouse *Oprm1* promoter, around -450 to -400 bp upstream of the open reading frame [172]. Phosphorylated Sp1 and Sp3 proteins bind to the morAP-2-like element to activate the *Oprm1* promoter [172]. In human neuroblastoma SK-N-SH and SH-SY5Y cells, chronic exposure to the selective MOR agonist, [D-Ala(2), N-Me-Phe(4), Gly(5)-ol] enkephalin (DAMGO) (0.1–10 µM, 48 h) results in enhanced binding of Sp1/Sp3 to the *OPRM1* promoter, which is attenuated by the pretreatment with the MOR antagonist, naloxone [173]. These data suggest that chronic opioid exposure may modulate Sp1/Sp3 binding at the MOR gene promoter to regulate its transcription.

The distal promoter region of mouse *Oprm1,* around −700 to −750 bp upstream of the open reading frame, contains multiple consensus binding sites of Sry-like high-mobility-group-box (SOX) factors [174,175]. In particular, overexpression of Sox18 can stimulate MOR gene transcription in human and murine neuroblastoma cells, suggesting that *Oprm1* is a downstream target of Sox18 protein [174]. In a cell culture model of human fetal brain-derived neural precursor cells (NPCs), prolonged cocaine treatment (1 µM, 3 days) suppresses the expression of SOX2, which is critical in the maintenance of embryonic and neural stem cells [127]. It is necessary in the future to investigate whether Sox proteins play roles in modulating *Oprm1* expression in animal models of SUD and identify which Sox family members are critically involved.

Nuclear factor kappa B (NF-κB) is another TF involved in SUD. NF-κB comprises an ubiquitous family of TFs that regulate various biological responses, such as immune responses and inflammation [176]. Blockade of NF-κB activity in mouse NAc inhibits cocaine reward [122]. The NF-κB binding sites are present in the promoter regions of both human and rodent MOR genes [6,177]. It has been confirmed that NF-ĸB binds to the promoter region of *OPRM1* and regulates its transcription [178]. It has also been shown that chronic cocaine treatment increases the protein levels of NF-κB subunits (e.g., p105/p50 and p65/Rel-A) in mouse NAc through epigenetic modification [120,122]. Moreover, prolonged treatment with morphine or DAMGO (1 nM–10 µM, 48 h) to human neuronal cells activates the promoter of NF-ĸB and thereby induces the expression of NF-ĸB [148]. The direct causal effect of NF-ĸB induction on *Oprm1* transcription in animal models of SUD has yet to be investigated.

∆FosB is a truncated splice variant of the FosB gene and heterodimerizes with Jun family proteins to form activator protein-1 (AP1), a TF that regulates its target genes, including *Oprm1*, via their promoters [179,180,181]. AP1 binding sites have also been identified at the human *OPRM1* promoter [182]. ΔFosB is induced in the striatum by virtually all drugs of abuse in animal models [183,184,185,186]. Enhanced ΔFosB levels are also reported in the NAc of human patients with a history of cocaine or opioid abuse [121]. Because ΔFosB is very stable, induced accumulation of ΔFosB may produce long-lasting effects on gene expression following drug withdrawal through AP1 binding to the promoters [187]. Of note, temporal alterations of histone acetylation and methylation were observed in concert with FosB gene induction [51,79,188,189], suggesting ΔFosB affects target gene expression via recruiting histone modifiers, such as HDACs, to the promoters of the target genes. In mice, the enrichment of H3K9me2 at the FosB promoter is sufficient to block ΔFosB expression and subsequent behavioral responses to cocaine [190], indicating that blockade of FosB gene expression using epigenetic approaches may have the potential to alleviate drug-seeking and taking behavior.

In humans, additional activating TFs have been identified for the *OPRM* gene such as PolyC-binding proteins [191], signal transducers and activators of transcription (STAT) [192], GATA-binding protein (GATA) [193], nuclear factor of activated T cells (NFAT) [194], Sp1/3 [195], yingyang-1 (YY1) [196] and PARP (poly-ADP ribose polymerase) [197]. As most of these findings were obtained from immune cells using in vitro systems, whether these TFs are expressed in different brain regions and whether they are directly involved in regulating *OPRM* expression in SUD merits further investigation.

There are also numerous repressive TFs for *Oprm1* expression. For example, PARP-1, an enzyme that repairs DNA and modifies a number of target proteins [198,199,200], binds to the double-stranded poly(C) element at the *Oprm1* promoter [201], and has been shown to act as either a positive or negative regulator of *Oprm1* expression [197,201]. A recent study shows that both acute and chronic cocaine exposure downregulate miR-124, which binds to the 3′UTR region of *Parp-1* mRNA and inhibits Parp-1 expression in mouse NAc, implicating its role in the effects of opioid administration [115]. Moreover, the 34-bp repressive cis-element, neuron-restrictive silencer element (NRSE), has been identified in the human *OPRM1* and mouse *Oprm1* genes, suggesting that conserved mechanisms mediated by NRSE suppress MOR expression in humans and rodents [115,202]. Other repressive TFs identified for MOR include: PU.1, a member of the ETS (E26 transformation-specific) family of TFs, binds to a 34-bp cis-acting element at the distal promoter of *Oprm1* [156]; octamer-1 (Oct-1) binds two sites at the proximal promoter (−121 to −100 and −42 to −22) of *Oprm1* [203]; two Sp3 isoforms bind to the 5′ UTR of *Oprm1* [204]; neuron-restrictive silencer factor (NRSF) binds to the NRSE of the *Oprm1* promoter [204]. The repressive effects of these TFs may be mediated by chromatin because treatment with Trichostatin A (TSA), an HDAC inhibitor, reverses the downregulation of *Oprm1* expression [156].

To date, most of the TFs that regulate *Oprm1* expression have been identified using cell lines or artificial gene reporter constructs. Therefore, the physiological relevance of most of these TFs on *Oprm1* expression remains to be validated in vivo and in different tissue types. Additionally, it is of great interest to explore whether drugs of abuse modulate the activities of these TFs, which ultimately change *Oprm1* gene expression.

#### 5.1.4. MOR Gene Expression, ncRNAs, and Drugs of Abuse

Multiple new reports indicate that ncRNAs are involved in *Oprm1* gene expression (Figure 3B). *Oprm1* has a long 3′ UTR, suggesting that this region may be targeted by miRNAs [205,206]. It has been shown that the binding of miR-339-3p or miR-23b to the 3′ UTR of *Oprm1* mRNA prevents it from interacting with polysomes, resulting in termination of *Oprm1* translation in mice [88,147]. In addition, both of these miRNAs inhibit murine MOR gene expression in vivo [207]. Interestingly, morphine treatment (10 nM–10 µM, 24 h) increases miR-339-3p and miR-23b expression in a concentration- and time-dependent manner in mouse neuroblastoma 2a (N2A) cells that stably express MOR [147]. Morphine-induced surges of these specific miRNAs repress mouse *Oprm1* expression by binding to the promoter. Moreover, Let-7 miRNAs bind to the 3′ UTR of the MOR gene and sequester *OPRM1* mRNA to P-bodies, which leads to translational repression of MORs in human SH-SY5Y cells [208]. Morphine treatment (24 or 48 h) to SH-SY5Y cells results in increased expression of all three Let-7 miRNAs, and subsequent repression of *OPRM1* expression [208]. These data suggest that drug exposure can alter the expression of the MOR gene by modulating specific miRNAs, which may explain the observed tolerance to morphine following prolonged or repeated exposure.

MRAK159688, a conserved lncRNA, is located in the nucleus and cytoplasm of neurons [152]. It binds to Swi-independent 3 transcription regulator family member A (Sin3a) and corepressor of REST (CoREST), both of which are key components of the repressor element-1 silencing transcription factor (REST) complex [209,210]. In rats that developed tolerance to repeated morphine treatment, MRAK159688 expression is markedly upregulated [152,211]. Downregulation of MRAK159688 partially reduces morphine tolerance and lessens morphine-induced hyperalgesia. This study supports the notion that MRAK159688 facilitates morphine tolerance by directly promoting REST-mediated repression of the MOR gene.

Opioid receptor genes also produce circRNA isoforms using canonical splice sites [212]. The *Oprm1* circRNAs (circOprm1) are conserved as they are present in the CNS of both rodents and humans. Currently, circRNAs have been detected for all members of the opioid receptor gene family [212]. Within *Oprm1* circRNA sequences, several miRNA binding sites are predicted, suggesting a potential role of *Oprm1* circRNA in sequestering miRNAand thereby modulating *Oprm1* levels. Of note, morphine treatment enhances specific *Oprm1* variant mRNAs in select brain regions [213]; however, it is unknown whether circOprm1 contributes to these changes. Since circRNAs are more stable than linear RNAs [214], circRNAs may potentially play a greater role in regulating *Oprm1* transcription and receptor function. Therefore, it is of future interest to further explore how chronic morphine treatment affects *Oprm1* expression by distinguishing the role of *Oprm1* circRNAs from that of mRNAs.

### 5.2. Epigenetic Modulation of the DOR Gene

#### 5.2.1. DOR Gene

The epigenetic regulation of *Oprd1* expression has not been extensively studied. A potential reason may be because DORs are not the primary target of opioids. Nevertheless, this receptor is involved in the reward pathway as well as anxiety-like behavior and contextual learning [215,216,217]. The promoter of the *Oprd1* gene is TATA-less and GC-rich. *Oprd1* has been mapped to mouse chromosome 4, contains two introns of 26 kb and 3 kb, and spans about 32 kb of chromosomal DNA [10,218]. Although *Oprd1* utilizes only one promoter, it contains two putative transcription initiation sites, located between −390 and −140 bp upstream from the open reading frame (Figure 2B) [10]. A single polyadenylation site is located 1.24 kb downstream from the stop codon [10]. There has been no alternative splicing reported for *Oprd1* mRNAs. In the P19 system, the *Oprd1* gene is constitutively active in undifferentiated cells but is silenced during neuronal differentiation, which is in stark contrast to the expression of *Oprm1* [219].

#### 5.2.2. OPRD1 SNPs and Drugs of Abuse

A promoter SNP rs569356, located 1968 bp upstream to the *OPRD1* transcription start site, has been identified as the only promotor SNP for the *OPRD1* gene [220]. This SNP is functional in cultured cells, as measured by luciferase reporter assays, and the substitute of a G for an A allele enhances *OPRD1* promoter activity, possibly through promoting TF binding [221]. Therefore, deep sequencing of the *OPRD1* promoter for rare variants is warranted in the future. Further, two coding sequence SNPs have been reported for the *OPRD1* gene: the synonymous rs2234918 in exon 3 and nonsynonymous rs1042114 in exon 1. The frequency of rs2234918, but not of rs1042114, is significantly higher in heroin-dependent patients, although this association is not consistently observed [222,223]. Moreover, the SNPs (rs2236861, rs2236857 and rs3766951) at intron 1 of *OPRD1* are highly associated in patients with heroin dependence [144]. To date, there is a lack of information regarding the effects of many identified SNPs on *OPRD1* transcriptional expression and receptor function.

#### 5.2.3. DOR Gene Expression and DNA Methylation

The GC-rich region of the mouse *Oprd1* promoter is subject to DNA methylation [224]. In mouse N2A cells, methylated CpG islands promote the binding of MeCP2, which recruits HDACs that remove histone acetylation in the *Oprd1* promoter, thus downregulating *Oprd1* expression [224,225]. The MeCp2-dependent repression of *Oprd1* can be reversed by the HDAC inhibitor TSA [225]. Conversely, when the promoter of *Oprd1* is demethylated, the promoter becomes more accessible to transcription factors and therefore *Oprd1* expression increases [225]. Human *OPRD1* gene expression is also regulated by promoter methylation. Hypermethylation at *OPRD1* promoter CpG sites is reported in blood samples of patients with Alzheimer’s disease [226]. To date, there is no study that focuses on the epigenetic modulation of *OPRD1*/*Oprd1* expression in animal models of SUD.

#### 5.2.4. DOR Gene Expression, Chromatin Modulating Factors, and Drugs of Abuse

*Oprd1* expression is modulated by histone modifications, such as histone acetylation. For example, treatment with nerve growth factor (NGF) to induce PC-12 cell differentiation into neurons produces a time-dependent induction of *Oprd1* expression [227,228]. It is further demonstrated that the NGF-induced increase in *Oprd1* expression is mediated by the activation of NF-κB, which can directly bind to the *Oprd1* promoter and stimulate promoter activity and gene expression [227,228]. Moreover, histone H3 K9 acetylation also contributes to an NGF-mediated increase in *Oprd1* expression. It is known that H3K9 is acetylated only when H3K9 is not methylated; NGF treatment leads to an approximately 40% reduction in the level of trimethylated H3K9 and a concomitant increase in acetylated H3K9, resulting in increased *Oprd1* expression [229].

Several other factors have been identified to activate *Oprd1* using reporter constructs in different cellular backgrounds. These factors are Ikaros [230,231], Sp1/Sp3 [232], E-twenty six 1 (Ets1) [233], upstream stimulatory factor (USF) [231,234], and AP2 [235]. Both in vivo and in vitro approaches demonstrated that increased binding activity of Ikaros (a zinc finger TF essential for immune cell development) at the Ikaros-binding site (−378 to −374) in the *Oprd1* promoter is required for transcription of the DOR gene in activated T cells [230]. Sp1 is ubiquitously expressed in mammalian cells and is essential for neuronal development [236]. The binding activity of Sp1 to its cis-acting element within the core promoter of *Oprd1* is cell-cycle and cell-type specific [232]. The mouse *Oprd1* promoter contains an E-box [233]; the binding of USF to the E-box is essential for constitutive expression of DOR in mouse neuronal NS20Y cells [234]. Further studies using NS20Y cells show that the Ets-1 binding site overlaps with the E-box and trans-activates the *Oprd1* promoter by coordinating with USF in specific DNA binding [233]. Another important TF is AP-2, which binds to the −157 bp region of the *Oprd1* promoter and activates the transcription of the DOR gene when mouse NG108-15 cells are differentiated [235].

There are only a few studies that investigate the modulation of *Oprd1* expression by TFs in the context of SUD. For instance, withdrawal from chronic alcohol consumption results in altered expression of several TFs previously implicated in DOR expression in the rat central nucleus of amygdala, including USF, AP2, and Ets1 [139]. Further, morphine treatment enhances AP-2 protein levels in the hippocampal postsynaptic density [150]. These dysregulated TFs could bind to the *Oprd1* promoter and modulate its expression. Lastly, morphine treatment (10–40 mg/kg, 4 days) increases the H3K9ac levels in mouse spinal dorsal horn [151]. Whether the increase in H3K9ac levels also occurs at the *Oprd1* promoter region has yet to be investigated and is of future interest.

### 5.3. Epigenetic Modulations of the KOR Gene

#### 5.3.1. KOR Gene

The KOR gene (*Oprk1*) has been mapped to mouse chromosome 1 [237,238,239]. Like *Oprm1*, mouse *Oprk1* utilizes two alternative promoters (P1 and P2) (Figure 2C). The P1 is present in various cultured cell lines and appears to be the primary promoter in animal tissues [240]. The P1 initiates transcription from a cluster of residues approximately 1kb upstream of the open reading frame. Since the alternative splicing can occur in intron 1, the P1 produces *Oprk1* mRNAs with one of two types of the 5′ UTR, and the mRNAs can span 4 exons. In contrast to the P1, the P2 is only active in certain brain tissues and at the later stages of differentiated neurons [241]. It is located within intron 1 and drives transcription from a specific residue at the −93 position, resulting in an mRNA product that spans only 3 exons [242]. Both P1 and P2 are TATA-less and GC-rich, similar to the gene promoters of other opioid receptors.

Molecular studies of *Oprk1* mRNA variants reveal that extensive post-transcriptional regulations occur throughout their UTRs, which control stability, location, and signaling specificity of the KOR protein. The *Oprk1* mRNA variants with distinct 5′ or 3′ UTRs have divergent half-lives [243]. In the P19 system, the most stable KOR mRNA variant has a half-life of 12 h, four hours longer than the second most stable variant under the same condition. Yet, the difference between these two variants is simply the insertion of 30 nucleotides in the 5′ UTR, generated from alternative splicing [244]. In addition, the mouse *Oprk1* gene can use two functional polyadenylating (PA) sites (PA1 and PA2) approximately 2 kb apart [245]. *Oprk1* transcripts that use these different PA sites display distinct mRNA stability, transcriptional and regulatory capacities demonstrated using in vitro reporter system. In vivo, each site is utilized in both neuronal tissues and cultured P19 cells but is differentially regulated depending on the stage of neuronal differentiation. This selective usage of PA sites in response to differentiation is partially due to the differential utilization of an adverse regulatory sequence adjacent to PA1 and/or an enhancer adjacent to PA2 [245]. Moreover, the stability of *Oprk1* mRNAs using PA2 is significantly greater than that using PA1, suggesting that the negative regulatory sequence adjacent to PA1 might be a regulatory target for an unknown miRNA(s). The physiological functions of these variants resulting from the alternative promoters, differential splicing, and distinct polyadenylation sites are unknown and should be thoroughly examined in the future.

#### 5.3.2. KOR Gene Expression, DNA Methylation, and Drugs of Abuse

DNA methylation contributes to the repression of the *Oprk1* gene. In an animal model of neuropathic pain, reduced *Oprk1* mRNA is accompanied by increased expression of DNMT3a (a de novo DNA methyltransferase) in rat DRG [246]. A blockade of DNMT3a expression elevates *Oprk1* levels in DRG cell culture, suggesting that *Oprk1* expression is modulated by DNA methylation. Chronic intermittent alcohol exposure increases the transcription and activity of DNMTs and global methylation and hydroxymethlation in rat NAc [133]. Interestingly, the mRNA levels of *Oprk1* are markedly increased by alcohol in the NAc of these animals, while the degree of methylation at the *Oprk1* promoter is not significantly reduced, suggesting that regulatory mechanisms other than DNA methylation may be involved. Furthermore, acute treatment with U50488, a KOR agonist, increases phosphorylation of MeCP2 protein at the serine residue 421 in mouse NAc but decreases it in the infralimbic and basolateral amygdala regions [247]. Phosphorylation of MeCP2 prevents the interaction between MeCP2 with the nuclear receptor co-repressor complex, thereby blocking its ability to repress transcription [248]. Therefore, altered phosphorylation of MeCP2 following acute stimulation of KORs may affect the ability of MeCP2 to suppress the transcription of *Oprk1* in a brain region-specific manner.

#### 5.3.3. KOR Gene Expression, Chromatin Modulating Factors, and Drugs of Abuse

The *Oprk1* gene is highly expressed in undifferentiated P19 cells, which is in contrast to the lower expression of the *Oprm1* gene [6]. However, during the differentiation of P19 cells into neurons, *Oprk1* expression initially falls but rises later on. This temporal pattern of *Oprk1* expression is related to the dynamic chromatin status of the promoter during P19 cell differentiation [249,250]. In undifferentiated P19 cells, the P1 promoter region is associated with the open chromatin conformation as evidenced by a low or more dynamic nucleosome occupancy, consistent with the constitutive activity of the P1 promoter [250]. Conversely, a higher-ordered chromatin structure was identified at the P1 region in differentiated P19 cells, which may block the accessibility of the P1 promoter and therefore decrease the transcription levels of *Oprk1*.

It appears that external stimuli can differentially regulate the mechanisms of *Oprk1* expression. For example, *Oprk1* expression can be induced by NGF at the P2 promoter, but not at the P1 promoter in P19 cells [249]. The activation of the NGF signaling pathway activates the downstream effector, AP2, which binds and activates the P2 promoter. The activated P2, during differentiation, is accompanied by increased H3K4 methylation, which is contrary to the presence of H3K9 methylation in undifferentiated cells [249]. Notably, exposure to stress has differential effects on the regulatory mechanisms of *Oprk1* expression. Repeated exposure to forced swimming, to induce stress in mice, increases *Oprk1* transcripts, preferentially controlled by the P1 promoter and terminated at polyadenylation site PA1, in specific mouse brain regions, including: the medial-prefrontal cortex, hippocampus, brainstem and sensorimotor cortex, but not in the amygdala and hypothalamus [251]. It is further demonstrated that the increased *Oprk1* expression results from reduced HDAC1 recruitment, increased acetylation of histone H4 (H4Ac), slightly reduced H3K4me2, and the recruitment of the transcription factor c-Myc on the P1 promoter. In contrast, there was no change in the H4Ac and H3K4me2 on the P2 promoter. It is also noted in the same study that the P2 promoter of *Oprk1* gene is active in specific brain regions, including the brainstem. These data indicate that stress stimuli can target different chromatin regions of *Oprk1* for transcriptional regulation, and this highlights the need to further investigate the *Oprk1* mRNA variants that result from these different regulatory mechanisms.

#### 5.3.4. KOR Gene Expression, TFs, and Drugs of Abuse

Various TFs have been identified that regulate mouse *Oprk1* expression. LMO4, a newly identified TF for *Oprk1* expression, can directly bind to the promoter of *Oprk1;* of interest, *Lmo4* gene knockout reduces *Oprk1* mRNAs, suggesting a direct regulatory relationship of Lmo4 in *Oprk1* transcription [252]. The same study also shows that reducing LMO4 proteins in rat basolateral amygdala suppresses alcohol drinking, which is likely, in part, mediated by a decrease in *Oprk1* gene expression as KORs in this brain region promote alcohol consumption. Besides LMO4, ∆FosB can modulate the signaling of KORs through the expression of dynorphin, a selective agonist for KORs. Direct binding of ∆FosB to the dynorphin promoter inhibits dynorphin expression in PC12 cells [253]. In addition, chronic morphine treatment leads to increased binding of ∆FosB at the dynorphin promoter in mouse NAc [149]. An epigenome-wide study of brain DNA methylation patterns in patients who died from acute opioid intoxication observed an increase in DNA methylation at a CpG site of the Netrin-1 gene [254]. Netrin-1 can stimulate the translation of KOR through Grb7, which is an RNA binding protein of KORs [255]. Therefore, DNA methylation at the Netrin-1 promoter likely leads to reduced translation of KORs.

There are several other TFs that are known to regulate *Oprk1* expression. c-Myc, Sp1 and AP2 positively regulate *Oprk1* transcription, while Ikaros (Ik) negatively regulates *Oprk1* transcription [249,256,257,258,259]. Nitric oxide (NO) is involved in transcriptional regulation of *Oprk1* via inactivating NF-κB [256]. NO downregulates c-Myc that binds and turns on the P1 promoter of *Oprk1* in P19 cells [257]. In a retinoic acid-induced P19 differentiation model, retinoic acid enhances Ik-1 expression, which recruits HDAC to intron 1 of the *Oprk1* promoter and represses its gene expression [259]. None of these TFs have been studied in animal models of SUD.

### 5.4. Epigenetic Modulation of the NOP Receptor Gene

#### 5.4.1. NOP Receptor Gene

NOP receptors are encoded by the opioid receptors like 1 gene (*Oprl1*). The human NOP receptor gene is located on chromosome 20 [154]. Like the other opioid receptors, the promoter of *Oprl1* is TATA-less and GC-rich [11,154]. The Human *OPRL1* gene is controlled by two alternate promoters, located approximately 10 kb apart (Figure 2D). It produces two transcripts with different start sites [154]. In contrast, rat *Oprl1* has two alternatively spliced isoforms, which are differentially expressed in tissues [260].

#### 5.4.2. NOP Receptor Gene Expression and Drugs of Abuse

There are only a few studies that focus on epigenetic modulation of *Oprl1*. Genome-wide methylation analysis of blood samples from monozygotic twins with or without alcohol dependence reveals a strong association of alcohol dependence with a differentially methylated site in the gene body of *OPRL1* [261]. In a cohort of European Americans that were alcohol-dependent with or without childhood adversity, analysis of blood DNA methylation levels in the promoter regions of targeted genes further shows that the *OPRL1* promoter is hypermethylated in patients regardless of childhood adversity experience [262]. In the same study, hypermethylation of the *OPRL1* promoter is also noted in drug naive patients that experienced childhood adversity, suggesting there was an epigenetic impact, due to the social stressors on the *OPRL1* gene expression. Moreover, a recent study, analyzing the methylation of two CpG sites located in the first intron of the *OPRL1* gene using blood samples from a pool of 14-year-old adolescent patients, reveals that low methylation at the first intron of the *OPRL1* gene in the blood is associated with early exposure to psychosocial stress and a high frequency of alcohol binge drinking. It is important to note that methylation at the promoter region is often associated with gene repression; however, methylation in the gene body does not necessarily cause gene silencing [263]. The direct causal relationship between stress, hypomethylation of the *Oprl1* gene body in the brain, and *Oprl1* expression is further validated in the brain of marchigian sardinian alcohol-preferring rats exposed to various types of stressors, including acute restraining, forced swimming, and sleep deprivation [264]. Stress notably induces hypomethylation of the *Oprl1* gene body in rat NAc, and a concomitant decrease in *Oprl1* gene expression and increased alcohol intake in these rats. Although there has been no study on the methylation of the *OPRL1* gene body and gene expression in the human brain, this type of epigenetic modulation of the *OPRL1* gene may contribute to the reduced *OPRL1* mRNA observed in the postmortem brain of patients with alcohol use disorder [137]. Collectively, these data suggest that drugs of abuse and environmental stressors can epigenetically modulate the expression of the NOP receptor gene.

Histone modifications also modulate *Oprl1* gene expression. Continuous cocaine administration via osmotic minipumps (50 mg/kg/day, 7 days) induces an increase in *Oprl1* expression in rat NAc but a decrease in the lateral caudate putamen [125]. Moreover, these changes are consistent with spatial alterations of H3K4me3 (an activation mark) and H3K27me3 (a repressive mark) in these two regions.

The human *OPRL1* promoter region contains binding sites for transcription factors, such as Sp1, AP2, EGR, Krox20, ETF, and CP1 or GCF (Figure 3) [11,154]. Multiple human genetic studies have identified high frequencies of *OPRL1* variants in patients with histories of substance use including opioids, alcohol, and psychostimulants [265,266,267]. Two *OPRL1* variants, rs6090041 and rs6090043, are significantly associated with a vulnerability to abuse opioids in Caucasians [265]. SNP rs6010718, is related to the development of alcoholism in a particular Scandinavian population [266]. These genetic variants of *OPRL1* may lead to altered epigenetic modulation of gene expression by changing the chromatin status and disrupting recruitment of TFs to the promoter.

### 5.5. Summary and Perspective

Drugs of abuse exert differential effects on transcriptional expression of opioid receptor genes, which may lead to changes in the function of opioid receptors and increased vulnerability for drug-seeking and taking behavior and other comorbid mental disorders. Although our knowledge of epigenetic modulation of opioid receptor genes has expanded in the past 30 years, there are still a few critical and poorly understood aspects that have yet to be studied. First, the expression of opioid receptor subtypes has been mostly measured using either the steady-state mRNA levels that are not determined at the transcriptional level or with reporter genes that lack regulatory 5′ or 3′ UTRs. It is important to note that different isoforms of opioid genes likely display distinct stability, RNA transport efficiency, local translation efficiency, and distribution patterns in the brain. Very little information is available on the expression and physiological function of different isoforms of opioid receptor genes in SUD. Second, most epigenetic mechanisms studied for opioid receptor genes are limited to modifications of DNA, post-translational modifications of histones, transcription factors, and ncRNAs. In addition to these mechanisms, different histone variants other than the canonical histones (H3, H4, H2A and H2B) and RNA modifications (e.g., N6-methyladenosine) may affect opioid receptor expression in SUD. N6-methyladenosine (m^6^A) was identified on *Ehmt2* mRNA, which codes for histone H3 K9 methyltransferase G9a and is involved in opioid receptor expression [268]. It would be interesting in the future to expand studies into these unexplored epigenetic mechanisms in the context of SUD. Third, recent data indicate cell type-specific epigenetic modulation of gene expression by drugs of abuse and its impact on SUD-related behavior. For example, chronic cocaine use increases *Hdac3* expression in *Drd1*- but not *Drd2*-containing cells in mouse NAc [269]. Notably, manipulation of HDAC3 activity in *Drd1*-containing neurons selectively regulates cocaine-associated memory formation and cocaine-seeking behavior. Future studies on cell type-dependent epigenetic modulations of opioid receptor genes by drugs of abuse are needed. Lastly, despite the fact that the promoters of all opioid receptors share some common effector profiles, such as DNA methylation and histone acetylation, the genes of the receptor subtypes undergo distinct forms of epigenetic regulation. Hence, it is critical to understand how HDAC or DNA methylation inhibitors would differentially affect the gene expression among the receptor subtypes.

## 6. Therapeutic Potentials of Drugs Epigenetically Targeting Opioid Receptor Gene Expression

In recent years, therapies targeting epigenetic processes have rapidly emerged in the management of human diseases. To date, small-molecular inhibitors targeting epigenetic writers, erasers, and readers have become available. A few of them are approved for clinical use. For example, the DNA methyltransferase (writer) inhibitors, 5-azaC (Vidaza) and 5-aza-dC (Dacogen) are clinically used for the treatment of myelodysplastic syndrome when stem cell therapy is not applicable [270]. Moreover, SAHA (suberoylanilide hydroxamic acid, known as Zolinza), was the first FDA-approved histone deacetylase (eraser) inhibitor for treating cutaneous T cell lymphoma [271]. It is utilized to restore sensitivity to chemotherapy in acute myeloid leukemia patients [272]. Bromodomain and extraterminal (BET) proteins are readers that regulate transcription by binding to acetylated histone lysine residues [273]. Recently, several structurally diverse BET inhibitors have been identified and are currently in clinical trials to treat cancer, diabetes, and other diseases [274]. Although most of these FDA-approved epigenetic drugs are limited in their intended use for hematological cancers [274], these drugs have been shown to be effective in alleviating symptoms of neurological diseases in rodents. For instance, systemic administration of 5-azaC produces a dose-dependent antidepressant-like effect, which is correlated with decreased DNA methylation and increased BDNF levels in rat hippocampus [275]. Further, early-life adversity increases DNA methylation. Treatment of dams exposed to maltreatment using zebularine, a DNA methylation inhibitor, reverses aberrant maternal behavior [276]. Additionally, SAHA is in clinical trials for treating neurological diseases such as Huntington’s disease and amyotrophic lateral sclerosis [274,277].

In animal models of SUD, manipulation of epigenetic modulators is effective in restoring dysregulated opioid receptor transcription and attenuating drug reward-related behavior. Therefore, drugs targeting aberrant epigenetic processes in SUD could serve as a new therapeutic avenue. Among all ORs, MORs are the target of most SUD-related research [278]. Extensive studies have shown that with chronic activation, MOR signaling is adapted in the brain with mostly upregulated MOR transcripts (Table 1). Such neuroadaptations weaken the drug effects and cause withdrawal symptoms to manifest [278]. Hence, downregulation of MOR gene expression may reduce drug reward, the motivation for drug seeking, and compulsive behavior. KORs also have well-recognized roles in driving SUD, while the functions of DORs are not as clear-cut. Decreased expression of KORs should restrict the dysphoric status related to stress and drug withdrawal, and prevent stress-induced relapse, although it may also promote reward. The BET inhibitors could block the reading of acetylated MOR and DOR promoters and reduce their gene expression, thereby reversing maladaptive transcriptional and behavioral responses to drugs of abuse. In fact, BET inhibitors have been shown to reduce cocaine- and opioid-seeking behaviors in rodents [279]. In addition, HDAC inhibitors have well-established roles in animal models of SUD [280,281,282].

Due to the complexity of epigenetic mechanisms in mammalian systems, there are discrepancies between human and animal experiments regarding the epigenetic influence on SUD [283]. Because drugs of abuse appear to affect multiple epigenetic targets, it is imperative to investigate whether it is beneficial to utilize combination drug therapy to target multiple aspects of an epigenetic process to control the transcription of opioid receptors. Future investigations are also necessary for validating the epigenetic modifications in the human substance-response pathways, developing site-specific epigenetic modifiers, and tracing patients’ drug-seeking and taking behavior after therapeutic interventions in a longitudinal manner.

## Figures and Tables

**Figure 1 ijms-23-11804-f001:**
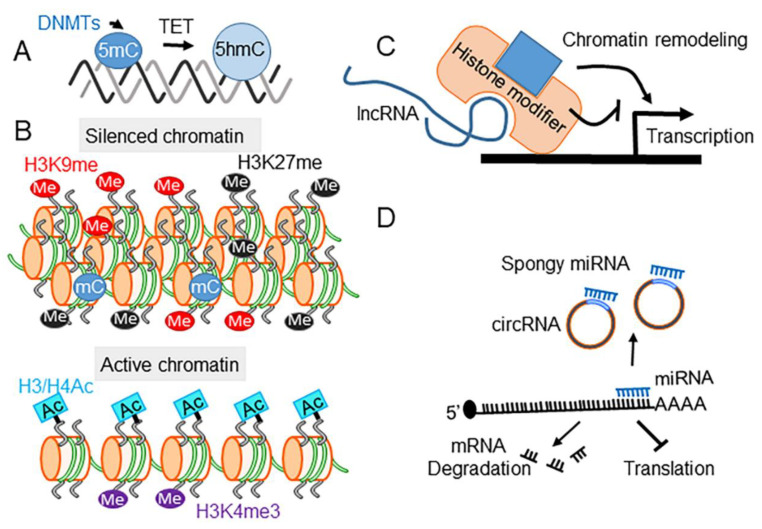
Schematic mechanisms of epigenetic mechanisms. (**A**) DNA methylation typically occurs at CpG dinucleotides (5mC), in which cytosines are methylated by DNA methyltransferases (DNMTs). The added methyl group(s) can be converted to hydroxymethyl (5hmC) by TET family proteins. Both 5mC and 5hmC are abundant in the brain. (**B**) Histone modifications mark the transcriptionally active or silenced chromatin states. Methylation and acetylation are two major studied histone modifications in drugs of abuse. The silenced chromatin is enriched with histone H3 lysine 9 (H3K9me) and 27 (H3K27me) methylations, whereas the active chromatin is covered by hyperacetylated histones and tri-methylated histone H3 lysine 4 (H3K4me3). (**C**) The long noncoding RNAs (lncRNAs) are likely involved in recruiting histone modifying enzymes that add or remove histone marks that control the local chromatin status, thereby, affecting transcription, either positively or negatively. (**D**) microRNAs (miRNAs) down-regulate mRNA levels either by mediating the degradation of mRNA or blocking translation. Circular RNAs (circRNAs) are a new class of ncRNA that may antagonize the function of miRNA by sponging and may also have gene regulation potency.

**Figure 2 ijms-23-11804-f002:**
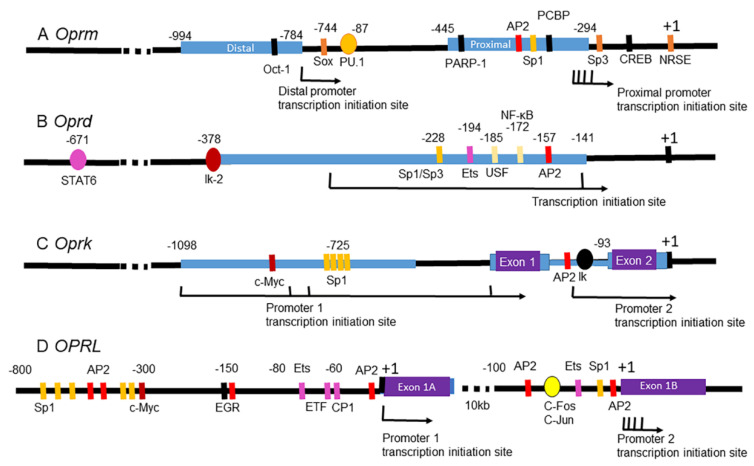
The regulatory DNA elements and corresponding TFs within the promoters of mouse *Oprm* (**A**), *Oprd* (**B**), *Oprk* (**C**), and human *OPRL* (**D**) genes. (**A**–**C**) Thick blue lines highlight validated promoters [6,154]. (**D**) Predicted regulatory elements and TF-binding sites at the human *OPRL* gene. Black arrows indicate reported transcription initiation sites from the promoters. (**A**–**D**) +1 shows the initiation codon of the open reading frame. Purple boxes are exons. Numbers above each map indicate the relative ends of validated promoter regions, regulatory sequences, or TF-binding sites, with respect to the +1 codon. Abbreviations: AP1, activator protein 1; AP2, activator protein 2; CREB, cyclic adenosine monophosphate (cAMP) response element binding protein; NF-κB, nuclear factor κB; Ets, E-twenty six; Ik, Ikaros; NRSE, neurorestrictive silencer element; Oct-1, octamer-1; PARP1, poly(ADP-ribose) polymerase 1; PCBP, poly C binding protein; PU.1, PU box binding; Sox, Sry-like high-mobility group box gene; Sp1, specificity protein 1; Sp3, specificity protein 3; STAT, signal transducers and activators of transcription; USF, upstream stimulatory factor; c-Myc, cellular Myelocytomatosis; EGR, early growth response protein; ETF, electron transfer flavoprotein; CP1, CCAAT transcription factor.

**Figure 3 ijms-23-11804-f003:**
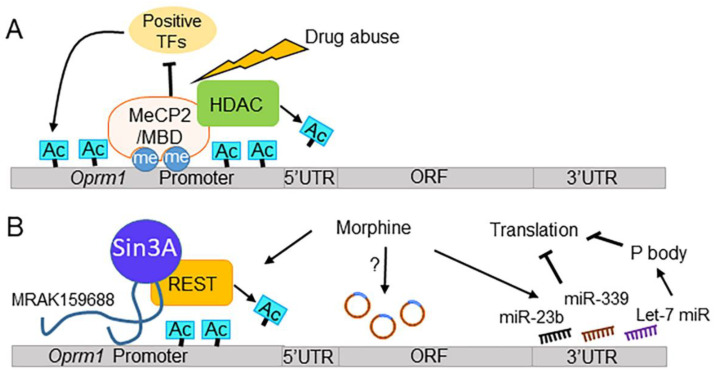
A hypothetical model of epigenetic suppression of *Oprm1* by drugs of abuse. (**A**) Potential cross-regulation between DNA methylation and histone modifications on *Oprm1*. The drugs of abuse enhance MeCP2 and other methyl-CpG-binding domain (MBD) proteins, which recognize DNA methylation and mark the promoter of *Oprm1*. This results in the recruitment of HDAC1, which removes the acetyl groups from the locus. The process suppresses the promoter from transcriptional activities; thus, *Oprm1* expression level is dampened. **(B**) ncRNA mediated suppression of *Oprm1*. Morphine treatment enhances the levels of MRAK159688 (lncRNA), miR-23b, miR-339 and Let-7 family miRNAs [152,165]. All these ncRNAs can recruit transcriptional co-repressors to suppress the transcription of *Oprm1.* Alternatively, they can limit *Oprm1* mRNA translation by directly binding to its 3′UTR or sequestering the mRNA at the processing body (P-body).

**Table 1 ijms-23-11804-t001:** Effects of drugs of abuse on mRNA levels of brain opioid receptors in vivo unless otherwise indicated.

Drug	Treatment	MOR	DOR	KOR	NOP	Related Findings	References
**Cocaine**	Acute treatment (sample collection within 20-120 min)	No change in *Oprm1* mRNA (rat)	N/A	No change in *Oprk1* mRNA (rat)	No change in *Oprl1* mRNA (rat)	↓ transcript levels of DNMT1, DNMT3a, TET1, and TET2 (mouse) ↓ miR-124 in NAc (mouse)	[66,114,115]
	Chronic treatment with short-term withdrawal period (30 min)	↑ *Oprm1* mRNA in NAc (rat)	No change in *Oprd1* mRNA (rat)	↓ *Oprk1* mRNA in SN, VTA, and NAc (rat)	N/A	↑ NF-κB subunits in NAc (mouse) ↑ ΔFosB in NAc (human) ↑ MEG3 (mouse HT22 cels in vitro) ↑ 5-mC levels in NAc (rat) ↑ DNMT transcripts and enzyme activity (mouse) ↓ H3K9me2 in NAc (mouse) ↓ PRMT6 and H3R2me2a in NAc (mouse) ↓ SOX2 (human NPCs) ↓ miR-124 in NAc (mouse)	[98,116,117,118,119,120,121,122]
	Chronic treatment with long-term withdrawal period (≥24 h)	↑ *Oprm1* mRNA in frontal cortex (rat) ↑ *Oprm1* mRNA in NAc (rat)	↑ *Oprd1* mRNA in CPu (rat) No change in *Oprd1* mRNA in NAc (rat)	↑ *Oprk1* mRNA in CPu (rat) No change in *Oprk1* mRNA in NAc (rat)	↑ *Oprl1* mRNA in NAc (rat) ↓ *Oprl1* mRNA in lateral CPu (rat)		[66,79,80,115,123,124,125,126,127]
**METH and MDMA**	Acute treatment (sample collection within 20–120 min)	No change in *Oprm1* mRNA (rat)	N/A	No change in *Oprk1* mRNA (rat)	↓ *Oprl1* mRNA in NAc by MDMA (rat)	↑ miR-181a-5p in NAc (rat) ↓ c-Fos in brain (male mouse)	[32,128,129]
	Chronic treatment with long-term withdrawal period (≥24 h)	↑ *Oprm1* mRNA (mouse)	No change in *Oprd1* mRNA (mouse)	↑ *Oprk1* mRNA in NAc (rat) No change in *Oprk1* mRNA (mouse)	↓ *Oprl1* mRNA in NAc by MDMA (rat)	↑ miR-143 in blood (human) ↓ miR-496-3p, miR-194-5p, and miR-200-3p in NAc (rat) ↑ c-Fos in amygdala (mouse) ↓ CREB in hippocampus (rat) ↓ BDNF in hippocampus (rat)	[32,89,128,130,131,132]
**Alcohol**	Chronic treatment with long-term withdrawal period (≥24 h)	↑ *Oprm1* mRNA in striatum (mouse), NAc and amygdala (rat)	↑ *Oprd1* mRNA in VTA and NAc (mouse)	↑ *Oprk1* mRNA in NAc (rat and mouse)	↓ *Oprl1* mRNA in NAc (rat)	↑ USF, AP2, and Ets1 in CeA (rat) ↑ DNMTs transcription and activity in NAc (rat) ↑ Global 5-mC and 5-hmC in NAc (rat) ↑ c-Fos in amygdala and hippocampus (rat)	[133,134,135,136,137,138,139,140]
** Opioids ** **Fentanyl**	Chronic treatment with long-term withdrawal period (≥24 h)	↑ *Oprm1* mRNA in PC12 cells (rat, in vitro)	No change in *Oprd1* mRNA in dorsal horn (mouse)	N/A	No change in *Oprl1* mRNA in SH-SY5Y cells (human *in vitro*)	↑ miR-339-3p in hippocampal neurons (mouse)	[88,141,142,143]
**Heroin**	Chronic treatment with long-term withdrawal period (≥24 h)	↓ *Oprm1* mRNA in NAc (rat)	N/A	N/A	N/A	↑ association of SNPs: rs2236861, rs2236857 and rs3766951) at intron 1 of *OPRD1* (human) ↑ H3K27ac in the striatum (human)	[72,144,145]
**Morphine**	Chronic treatment with a long-term withdrawal period (≥24 h)	No change in *Oprm1* mRNA in PC12 cells (rat) ↓ *Oprm1* mRNA in PAG (rat in vitro) ↓ *Oprm1* mRNA in SH-SY5Y cells (human in vitro)	↑ *Oprd1* mRNA in the spinal cord (rat) ↓ *Oprd1* mRNA in PAG (rat)	↑ *Oprk1* mRNA in the spinal cord (rat)	↓ *Oprl1* mRNA in SH-SY5Y cells (human in vitro)	↑ miR-339-3p in hippocampal neurons (mouse) ↑ miR-23b in N2A cells (mouse in vitro) ↑ MRAK159688 expression (rat) ↑ AP-2 in the hippocampal postsynaptic density (mouse) ↑ H3K27ac in spinal dorsal horn (mouse) ↑ ∆FosB binding at dynorphin promoter in NAc (mouse) ↑ NF-κB expression in neuronal cells (human in vitro)	[88,141,142,146,147,148,149,150,151,152]
**Oxycodone**	Acute treatment (sample collection within 20–120 min)	↑ *Oprm1* mRNA in the spinal cord (rat)	No change in *Oprd1* mRNA in the spinal cord (rat)	No change in *Oprk1* mRNA in the spinal cord (rat)	N/A	N/A	[153]

Note: CPu, caudate putamen; NAc, nucleus accumbens; SN; substantia nigra; VTA, ventral tegmental area; CeA, central nucleus of the amygdala. ↑ indicates upregulation; ↓ indicates downregulation.

## Data Availability

This article does not report any data.

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
