# Peer review of "Epigenetic Modulation of Opioid Receptors by Drugs of Abuse"

_ijms, 2022, doi:10.3390/ijms231911804_

Round 1
Reviewer 1 Report
The review by Reid et al. is timely, as there has been increased focused on the role of epigenetics in substance use disorder in the last ten years. Given the importance for opioid signaling in addiction, discussion of epigenetic mechanisms controlling opioid receptor expression should be of wide interest. The authors cover a large number of topics including descriptions of various types of epigenetic modifications, the structure and function of the four types of opioid receptors, epigenetic changes induced by drugs of abuse, as well as transcription factors known to modulate opioid receptor transcription. While the authors extensively cite the literature, the review also seems a bit unfocused in places. Although the article is entitled “epigenetic modulation of opioid receptors by drugs of abuse” discussion of this specific literature is rather brief, and the focus on this topic is somewhat masked by the extensive discussion of topics outside this focus. Additionally, many sections would benefit from summaries and “take-home” statements as paragraphs read as a list of findings without providing an interpretation of what the combined data indicate as a whole. Given that the authors cover an extensive literature, incorporating tables may help to summarize the specific findings while giving the authors more space to interpret those findings and put them into context for the reader.
In summary, this is a topic worthy of review and the authors have extensively surveyed the literature. The authors may consider the points below to help focus the review and ensure that their main thesis is evident to scientists outside the field.
1. Ensure that each section has a thesis and that this clearly supported and summarized.
2. Consider using tables to convey details in data-heavy sections so that there is space to tie the studies together in a cohesive narrative.
3. While the authors extensively discuss mechanisms that can alter opioid receptor gene expression, they do not provide details on how drugs of abuse alter expression of the 4 types of opioid receptors (lines 95-97, a few papers are cited but there is no description on how expression is changed and which opioid receptors are being discussed). This seems like an important point that should be discussed (again a table could be used if the literature it too extensive to discuss). Within this point, it should also be discussed whether such changes in expression are long-lasting, as this helps to justify looking into epigenetic mechanisms.
4. Consider removing section 6 as it is not well-integrated with the rest of the review and as currently written does not offer any specific insight into the benefit of epigenetic modification of opioid receptor genes for substance use disorder.
5. The review is generally well-written but would benefit from copy-editing (e.g., subject-verb agreement, tense, etc.).
Author Response
In response to insightful comments by the reviewers, we have revised the paper by adding a table and modifying the text. We are grateful for the improvements and thank our reviewers for their useful comments. Here is a point-by-point response to reviewer #1's comments (italicized):
Ensure that each section has a thesis and that this clearly supported and summarized.
We have revised the text to ensure that each section has a theme and is well-supported by the summarized content.
Consider using tables to convey details in data-heavy sections so that there is space to tie the studies together in a cohesive narrative.
We are grateful for this suggestion and we have added a table in section 5 to summarize how drugs of abuse modulate (increase or decrease) four opioid receptor expressions and related alterations of epigenetic events.
While the authors extensively discuss mechanisms that can alter opioid receptor gene expression, they do not provide details on how drugs of abuse alter expression of the 4 types of opioid receptors (lines 95-97, a few papers are cited but there is no description on how expression is changed and which opioid receptors are being discussed). This seems like an important point that should be discussed (again a table could be used if the literature it too extensive to discuss). Within this point, it should also be discussed whether such changes in expression are long-lasting, as this helps to justify looking into epigenetic mechanisms.
This is a great suggestion. We have now added a table in section 5 to summarize how drugs of abuse modulate (increase or decrease) four opioid receptor expressions and related alterations of epigenetic events. Short-term (acute) or long-term (chronic) effects are included in table 1.
Consider removing section 6 as it is not well-integrated with the rest of the review and as currently written does not offer any specific insight into the benefit of epigenetic modification of opioid receptor genes for substance use disorder.
We appreciate the reviewer's suggestion. We have modified section 6 by adding rationale, hypothesis, and possible targets of epigenetic-based medication development for the treatment of SUD.
The review is generally well-written but would benefit from copy-editing (e.g., subject-verb agreement, tense, etc.).
We have followed with reviewer's comment and revised the text for correct subject-verb agreement, tense, etc.
We believe that we have addressed all the points raised by the reviewers. We also hope that these changes are satisfactory.
Reviewer 2 Report
Overall, it is a well-written article, summarizing recent progress in research regarding epigenetic modulation of opioid receptors by drugs of abuse. I have only a few comments and suggestions for the authors to further improve this manuscript.
1) The section 4 looks too long and off-topic. The authors may consider using a table to summarize the major findings on this topic and then shorten the text by ~50%.
2) Section 5: A major concern is regarding the molecular mechanisms underlying epigenetic modulations of opioid receptors by drugs of abuse, which is not well addressed. The authors may add more discussion and use one to two more diagrams illustrating how psychostimulants (cocaine, methamphetamine) and opioids (morphine, fentanyl) alter (increase or decrease) opioid receptor expression via DNA methylation, histone modification or ncRNA. It is well known that almost all drugs of abuse may activate the mesolimbic dopamine (DA) system and elevate extracellular DA level in the NAc. This reviewer wonders whether a DA-dependent mechanism underlies drug-induced epigenetic modulation of opioid receptors. If not DA, what are the other possible mechanisms underlying those epigenetic modulations of opioid receptors
3) Section 5: The authors may consider to add 1 or 2 tables to summarize literature reports, indicating i) how drugs of abuse modulate (increase or decrease) opioid receptor expression; ii) what is the function significance of the epigenetic modulations on opioid receptor signaling and opioid-taking and opioid-seeking behavior.
4) Section 6: The authors may tell us more about epigenetic-based medication development for the treatment of OUDs - rationale, hypothesis, possible targets, etc.
Author Response
We appreciate your kind help in reviewing our manuscript. In response to insightful comments by reviewer #2, we have revised the paper by adding a table and modifying the text. We are also grateful for the improvements and thank our reviewers for their useful comments.
Below is our detailed response to specific comments by reviewer #2, which are indicated in italicize:
The section 4 looks too long and off-topic. The authors may consider using a table to summarize the major findings on this topic and then shorten the text by ~50%.
Considering this review article is submitted to Molecular Pathology, Diagnostics, and Therapeutics section, we think the audience would appreciate a more detailed introduction to the molecular mechanisms of epigenetics. We do have a diagram (Figure 1) to illustrate the major points in this section.
2) Section 5: A major concern is regarding the molecular mechanisms underlying epigenetic modulations of opioid receptors by drugs of abuse, which is not well addressed. The authors may add more discussion and use one to two more diagrams illustrating how psychostimulants (cocaine, methamphetamine) and opioids (morphine, fentanyl) alter (increase or decrease) opioid receptor expression via DNA methylation, histone modification or ncRNA. It is well known that almost all drugs of abuse may activate the mesolimbic dopamine (DA) system and elevate extracellular DA level in the NAc. This reviewer wonders whether a DA-dependent mechanism underlies drug-induced epigenetic modulation of opioid receptors. If not DA, what are the other possible mechanisms underlying those epigenetic modulations of opioid receptors
We appreciate this important comment. We have added a table in section 5 to summarize how drugs of abuse modulate (increase or decrease) four opioid receptor expressions and related alterations of epigenetic events.
Indeed, a DA-dependent mechanism underlies drug-induced epigenetic modulation of opioid receptors. However, numerous articles have reviewed this topic. We, therefore, just briefly discussed this topic in the manuscript.
3) Section 5: The authors may consider to add 1 or 2 tables to summarize literature reports, indicating i) how drugs of abuse modulate (increase or decrease) opioid receptor expression; ii) what is the function significance of the epigenetic modulations on opioid receptor signaling and opioid-taking and opioid-seeking behavior.
We have followed the reviewer's suggestion and added Table 1 to summarize the literature reports.
4) Section 6: The authors may tell us more about epigenetic-based medication development for the treatment of OUDs - rationale, hypothesis, possible targets, etc.
We are grateful for this insightful comment and we have modified section 6 by adding rationale, hypothesis, and possible targets of epigenetic-based medication development for the treatment of SUD.
We believe that we have addressed all the points raised by the reviewer. We also hope that these changes are satisfactory.